# GSR Deficiency Exacerbates Oxidative Stress and Promotes Pulmonary Fibrosis

**DOI:** 10.3390/biom15071050

**Published:** 2025-07-20

**Authors:** Wenyu Zhao, Hehe Cao, Wenbo Xu, Yudi Duan, Yulong Gan, Shuang Huang, Ying Cao, Siqi Long, Yingying Zhang, Guoying Yu, Lan Wang

**Affiliations:** State Key Laboratory of Cell Differentiation and Regulation, Henan International Joint Laboratory of Pulmonary Fibrosis, Henan Center for Outstanding Overseas Scientists of Organ Fibrosis, Pingyuan Laboratory, College of Life Science, Henan Normal University, Xinxiang 453007, China; 2004183088@stu.htu.edu.cn (W.Z.); 2304224042@stu.htu.edu.cn (H.C.); 2314114069@stu.htu.edu.cn (W.X.); 2104183035@stu.htu.edu.cn (Y.D.); 2304183086@stu.htu.edu.cn (Y.G.); 19040174126@stu.htu.edu.cn (S.H.); 2304183002@stu.htu.edu.cn (Y.C.); 2304183043@stu.htu.edu.cn (S.L.); 19040174127@stu.htu.edu.cn (Y.Z.)

**Keywords:** idiopathic pulmonary fibrosis, type II alveolar epithelial cell, fibroblast, GSR, oxidative stress, ROS, aging

## Abstract

Idiopathic pulmonary fibrosis (IPF) is a progressive and fatal lung disorder characterized by excessive scarring of lung tissue, predominantly affecting middle-aged and elderly populations. Oxidative stress plays a pivotal role in the pathogenesis of pulmonary fibrosis, disrupting redox homeostasis and driving fibrotic progression. Glutathione reductase (GSR), a key antioxidant enzyme, is essential for maintaining cellular glutathione (GSH) levels and mitigating oxidative damage. However, the specific involvement of GSR in IPF remains poorly understood. This study found that GSR levels were downregulated in IPF patients and mice treated with bleomycin (BLM). GSR knockdown enhanced epithelial-to-mesenchymal transition (EMT) in A549 cells and promoted the activation of MRC5 cells. Additionally, GSR depletion promoted cellular migration and senescence in both A549 and MRC5 cells. Mechanistically, silencing GSR in A549 and MRC5 cells led to a marked reduction in intracellular GSH levels, resulting in elevated reactive oxygen species (ROS) accumulation, thereby promoting the activation of the TGF-β/Smad2 signaling pathway. In conclusion, our findings demonstrate that GSR deficiency aggravates pulmonary fibrosis by impairing antioxidant defense mechanisms, promoting EMT, and activating fibroblasts through the TGF-β/Smad2 signaling. These findings suggest that GSR may be essential in reducing the fibrotic progression of IPF.

## 1. Introduction

Idiopathic pulmonary fibrosis (IPF) pathogenesis encompasses alveolar and fibrotic remodeling, inflammation, and eventual loss of lung architecture [1], resulting in progressive loss of pulmonary function, respiratory failure, and death, often within 5 years of diagnosis [2,3,4,5]. The widely accepted understanding of IPF involves chronic alveolar damage and impaired restoration of the respiratory epithelium by type II alveolar epithelial cells (AECIIs). Pathological hallmarks typically encompass excessive accumulation of extracellular matrix (ECM), fibroblastic abnormalities, and the progressive substitution of normal tissue in the lung parenchyma with altered ECM [6]. Severe cases can cause hypoxia, acidosis, loss of labor, and death. Pirfenidone and Nidanib are approved treatments for IPF [7]. The therapeutic effects of pirfenidone and Nidanib on IPF patients have been shown to slow down the progression of the disease [8]. These treatments have side effects and are not a complete cure for IPF, which remains incurable despite more treatment options.

Glutathione reductase (GSR) is a key enzyme in animal redox systems and is central to cellular antioxidant defenses [9]. GSR uses nicotinamide adenine dinucleotide phosphate (NADPH) to convert oxidized glutathione (GSSG) back to reduced glutathione (GSH) [10]. Oxidized glutathione is transformed into reduced glutathione, which maintains the redox state of glutathione in cells. GSH is the main antioxidant in mammalian cells [11], in almost all cellular metabolism and the cell cycle [12]. It can neutralize hydroxyl radicals, lipid peroxide radicals, hypochlorous acid, reactive oxygen species, and various other electrophilic agents [13]. Cells are protected from oxidative damage by glutathione, a key component of both intracellular and extracellular antioxidant defenses. Bronchoalveolar lavage (BAL) analysis showed that IPF patients have reduced GSH in their epithelial lining fluid. Additionally, healthy individuals have about four times more GSH in their sputum than IPF patients [14].

The human lungs are vulnerable to oxidative stress [15], which is believed to be closely related to pulmonary fibrosis [16]. In lung tissue, where oxygen levels are plentiful, numerous mitochondria reside. These organelles utilize oxygen and generate reactive oxygen species (ROS) as a by-product [17]. In addition, bleomycin (BLM) produces large amounts of ROS when exposed to oxygen [18]. ROS can damage pulmonary epithelial cells [19] and stimulate epithelial cells to release high levels of cytokines, leading to the process of epithelial-to-mesenchymal transition (EMT) [20], which is considered to be an important process in the development of pulmonary fibrosis. EMT is a process where epithelial cells lose junctions, reorganize their cytoskeleton, and adopt mesenchymal characteristics like invasiveness, migration, and increased ECM production [21]. Among pharmacological ROS scavengers, N-acetyl-L-cysteine (NAC) serves as a potent synthetic antioxidant that attenuates ROS-dependent apoptotic pathways [22]. NAC functions as a cysteine prodrug that enhances glutathione synthesis, conferring dual antioxidant actions: direct thiol-mediated radical neutralization and indirect glutathione-dependent ROS detoxification [23]. ROS scavengers, particularly NAC, are frequently employed to verify that ROS is involved in drug-induced apoptosis [24].

Transforming Growth Factor-beta (TGF-β) reduces mitochondrial electron transport efficiency in epithelial cells, decreasing membrane potential and increasing mitochondrial Reactive Oxygen Species (ROS). High ROS levels activate TGF-β, promoting fibroblast accumulation [25,26,27,28]. Recurrent injury to epithelial cells triggers wound repair processes, which result in the secretion of cytokines. These cytokines then stimulate fibroblast multiplication and enhance collagen synthesis [29,30]. The TGF-β pathway is the main driver of fibroblast differentiation, triggering processes like increased collagen production, matrix deposition, ECM formation, and α-smooth muscle actin (α-SMA) expression [31,32]. These actions collectively expedite the progression of pulmonary fibrosis.

Our research revealed that the downregulation of GSR results in the EMT progression of AECIIs and fibroblast activation. Genetic perturbation of GSR enhanced migratory capacity and induced cellular senescence in both AECIIs and fibroblasts. Mechanistically, GSR silencing depleted intracellular GSH, triggering ROS accumulation. This ROS surge subsequently activated the TGF-β/Smad2 signaling pathway. These findings suggest that GSR may be essential in reducing the fibrotic progression of IPF.

## 2. Materials and Methods

### 2.1. Collection of Lung Tissue Samples from Mice

All animal procedures were approved and performed according to the guidelines of the Institutional Animal Care and Use Committee at Henan Normal University (IACUC, SMKX-2118BS1018, No. 2021-03-08).

In this study, 6–8 week-old male C57BL/6 mice, weighing 18–23 g, were used to create a pulmonary fibrosis model with a single 3 mg/kg BLM intratracheal dose under anesthesia, developing the model after 14 days. Anesthetized mice were then dissected to collect lung tissue, blood, and alveolar lavage fluid using surgical tools. Two whole lungs per group were randomly chosen, trimmed, and preserved in 4% paraformaldehyde.

### 2.2. Isolation of Primary Human Pulmonary Fibroblasts (PHLFs) and Primary Mouse Pulmonary Fibroblasts (PMLFs)

#### 2.2.1. Isolation of PHLFs

The isolation and culture procedure of PHLFs is described as follows: Briefly, lung tissues obtained from healthy donors were cut into small pieces with a diameter of approximately 1 mm. These pieces were then submerged in Dulbecco’s modified Eagle’s medium (Gibco, Life Technologies Corporation, Carlsbad, CA, USA) supplemented with collagenase II (2 mg/mL), trypsin (2.5 mg/mL), DNase I (2 mg/mL), penicillin (100 U/mL), and streptomycin (100 μg/mL). The mixture was incubated at 37 °C on a shaker for 12 h. After that, the tissues were rinsed with Dulbecco’s phosphate-buffered saline. Next, the cells were centrifuged at room temperature at 500× *g* for 5 min. Subsequently, they were cultured in high-glucose complete medium (containing 10% fetal bovine serum, 1% penicillin (100 U/mL), and streptomycin (100 μg/mL) (Gibco, Life Technologies Corporation, California, USA)) at 37 °C in an environment with 5% CO_2_. The clinical research was approved by the Medical Research Ethics Committee of Henan Provincial Chest Hospital (No. 2020–03–06).

#### 2.2.2. Isolation of PMLFs

The detailed operation procedure is outlined briefly as follows: 3-day-old mice are humanely euthanized using professional anesthetic agents. Subsequently, they are thoroughly cleaned with 75% ethanol. Throughout the operation, strict aseptic operation guidelines must be followed.

The lung tissue of the mice is harvested and rinsed twice with sterile PBS. Next, the lung tissue is accurately cut into tissue blocks sized 1 cubic millimeter. Once the cutting is finished, the tissue blocks are centrifuged at 600× *g* for 5 min. After centrifugation, the sediment is carefully resuspended in DMEM medium supplemented with 10% fetal bovine serum. Then, it is transferred to a 10 cm culture dish and cultured in an environment maintained at 37 °C.

After the tissue blocks go through standard procedures, including digestion and filtration, they are cultured continuously for 4–5 days. At this stage, the adherent fibroblasts are collected and can be utilized for subsequent subculture or other relevant experiments.

### 2.3. Cell Culture

The cell experiments were carried out by using MRC5 (human embryonic lung fibroblasts) and A549 (human lung adenocarcinoma cells) from the American Type Culture Collection (ATCC). Among them, the A549 cell culture medium is a complete medium prepared from the DMEM-F12 basic medium. The MRC5 cell culture medium is a complete medium prepared from DMEM basic medium. Besides, the PHLFs are cultured by high-glucose complete medium, and the PMLFs are cultured by DMEM medium. The medium contained 10% fetal bovine serum (FBS), 100 units/mL penicillin, and 100 g/mL streptomycin. The media were changed every three days. Cells were cultured in cell culture bottles or cell culture dishes in a humidity-saturated cell incubator with a temperature of 37 °C and a CO_2_ content of 5%.

### 2.4. Hematoxylin and Eosin (H&E), Masson’s Trichrome Staining, and Immunohistochemistry (IHC)

Lung tissues were first fixed in 4% paraformaldehyde and then embedded in paraffin to make 5-μm tissue sections. These tissue sections underwent a process of deparaffinization and rehydration using xylene and ethanol of various concentrations. Hematoxylin and eosin (H&E) staining as well as Masson’s trichrome staining were carried out following the instructions of the kit (Solarbio, Beijing, China). Subsequently, histopathological changes were examined under a microscope.

Regarding immunohistochemistry (IHC), the immunohistochemical staining steps were as follows: The lung sections were first dewaxed until transparent and then treated with 3% H_2_O_2_ by dropwise addition for 10 min. After being rinsed with distilled water, the slides were immersed in a 0.01% sodium citrate solution and heated in a microwave until boiling. Next, 50% goat serum was used for blocking for 30 min, followed by overnight incubation with the primary antibody. On the next day, the horseradish peroxidase-labeled antibody was incubated for 30 min at 37 °C, and then DAB chromogen was used for development. After staining the nuclei with hematoxylin for 5 min, the sections were rinsed with running water for 10 min. Finally, after being sealed with neutral balsam, five visual fields were randomly chosen from each section for evaluation.

### 2.5. Transwell and Cell Scratch Test

Cells were resuspended in serum-free DMEM-F12 and seeded in a transwell insert without BD matrix glue. The lower chamber was filled with DMEM-F12 plus 10% FBS. Migrated cells were counted in five random fields, stained, imaged, and analyzed. Cells were cultured to confluence in six-well plates, scratched with a pipette tip, and cell migration was monitored under a microscope at 0 and 48 h post-injury.

### 2.6. Reverse Transcription qPCR (RT-qPCR)

Total RNAs isolated from human lung tissue samples were extracted with TRIzol Reagent (Life Technologies, USA). In the RT-qPCR experiment, TRIzol Reagent was used to extract total RNA from cultured cells that had undergone different treatments. Subsequently, MLV Reverse Transcriptase (M1708, Promega, Fitchburg, WI, USA) was employed to synthesize cDNA. RT-qPCR was conducted using a SYBR green kit (11201ES03, Yeasen, Shanghai, China) according to the manufacturer’s instructions. Each sample and experiment was tested in triplicate. Delta CT values of the target gene were normalized to GAPDH. The data were evaluated by the 2^−ΔΔCt^ method. GSR: F-TATGTGAGCCGCCTGAATGCCA, R-CACTGACCTCTATTGTGGGCTTG. Gsr: F-GTTTACCGCTCCACACATCCTG, R-GCTGAAAGAAGCCATCACTGGTG. GAPDH: F:GTCTCCTCTGACTTCAACAGCG, R: ACCACCCTGTTGCTGTAGCCAA. GAPDH: F: GAGAGGCCCTATCCCAACTC, R: TCAAGAGAGTAGGGAGGGCT.

### 2.7. Western Blot

Protein was obtained from cell lysate in lysis buffer and used for protein quantification. Protein was separated by sodium dodecyl sulfate-polyacrylamide gel electrophoresis, transferred to a polyvinylidene fluoride membrane, and detected by the specific antibody. The chemiluminescence kit purchased from Thermo Fisher Scientific Shier Technology Company was used to detect protein. The imager station captured the images (Odyssey Software Version 5.2, LI-COR Biosciences, Lincoln, NE, USA). The following antibodies were used: GSR (18257-1-AP, Proteintech, Wuhan, China), β-actin (AF7018, Affinity, Changzhou, China), GAPDH (AF7021, Affinity, China), α-SMA (ab124964, Abcam, Cambridge, UK), vimentin (10366-1-AP, Proteintech, Wuhan, China), E-cadherin (14472S, Cell Signaling Technology, Boston, MA, USA), N-cadherin (13116T, Cell Signaling Technology, Boston, USA), COL1A1 (14695-1-AP, Proteintech, Wuhan, China), FN1 (15613-1-AP, Proteintech, Wuhan, China), Smad2 (83841-5-RR, Proteintech, Wuhan, China), p-Smad2 (44-244G, Thermo, Waltham, MA, USA), and TGF-β (ab9758, Abcam, Cambridge, UK).

### 2.8. Reactive Oxygen Detection

The ROS detection was based on the Reactive Oxygen Species Assay Kit (CA1410, Solarbio, Beijing, China). For this, 3000 cells were inoculated in each well of a 96-well plate, and the cells were treated normally after they adhered to the wall. After the cell treatment time was over, the culture medium in the hole was sucked and discarded, and PBS was added to wash it once. Then, a DCFH-DA working solution was prepared. The ratio of DCFH-DA to serum-free medium was 1:1000. Then, 0.1 mL of the working solution was added to each well of a 96-well plate and incubated in an incubator at 37 °C for 20 min. Then, it was washed with serum-free culture solution three times for one minute each time. After adding 100 µL PBS into each well, it was observed, and photos were taken with an inverted fluorescence microscope, or the light absorption value in a fluorescent microplate reader was read.

### 2.9. Determination of Glutathione Content and Lipid Peroxide Content

GSH content was determined by the reduced GSH detection kit (BC1170, Solarbio, Beijing, China), and MDA content was determined by the lipid oxidation detection kit (S0131S, Beyotime, Shanghai, China), and the operation was carried out according to the instructions of the kits.

### 2.10. β-Galactosidase Staining

The β-galactosidase staining kit (G1580, Solarbio, Beijing, China) was used to detect cell senescence. The culture medium from the treated cells was discarded in the six-well plate and washed twice with PBS buffer. Then, 1 mL of β-galactosidase staining fixative was added to each well and fixed at room temperature for 0.25 h. The cell fixative was sucked with a pipette gun, and two milliliters of PBS buffer was added to each hole to wash the cells three times, each time for three minutes. After discarding PBS, 1 mL of dyeing working solution prepared in advance was added to each well. The six-well plate was sealed with plastic wrap to prevent evaporation and incubated overnight in a constant-temperature incubator at 37 °C. After staying overnight, the cells with positive β-galactosidase staining were observed and counted under a microscope.

### 2.11. TGF-β ELISA Experiment

The concentration of TGF-β in the cell supernatant was determined by the Human TGFBI ELISA KIT (SEKH-0316, Solarbio, Beijing, China), and the operation was carried out according to the instructions of the kit.

### 2.12. Cell Co-Culture System

The conditioned medium was collected after interfering with GSR72 in A549 cells for 72 h. The MRC5 cells were cultured in the conditioned medium for 72 h. The protein of MRC5 cells was extracted after cell treatment, and the protein expression was detected by Western blot.

### 2.13. Gene Expression Data

The analysis of GSR gene expression was conducted using the GSE35145 data from the GEO database. The ggplot2 package (version 3.5.1) in R (version 4.4.2) was employed to draw the box plot, which intuitively presented the distribution characteristics of the data. The ggsignif package (version 0.6.4) was utilized to perform statistical tests on the data between groups. The “test” parameter was set as “wilcox.test” to execute the Wilcoxon test.

### 2.14. Statistics and Analysis

The outcomes of the control and experimental groups were analyzed using GraphPad software version 9.0. The Shapiro–Wilk normality test was employed to check for normal distribution. When the sample data did not follow a normal distribution, the Mann–Whitney U test was utilized to analyze the results for comparisons between two groups. In the case of normal distribution of sample data for two groups, the unpaired Student’s *t*-test was used for such comparisons. We used two-way analysis of variance (ANOVA) for normally distributed and homoscedastic data for pairwise comparisons of three or more groups, followed by a Bonferroni post hoc test. Data are presented as mean ± SD and were considered statistically significant at *p* < 0.05.

## 3. Results

### 3.1. Downregulation of GSR Expression in Pulmonary Fibrosis

To investigate the potential role of GSR in IPF, we analyzed a publicly available dataset of IPF samples and discovered a significant downregulation of GSR mRNA levels in lung tissues from IPF patients (Figure 1A). IHC results also showed that GSR is significantly downregulated in lung tissues from IPF patients compared to healthy controls (Figure 1B). We then investigated the role of GSR in A549 cells by exposing them to BLM for 48 h to create a cellular injury model. A significant reduction in the protein and mRNA levels of GSR in the BLM-treated group compared to controls was demonstrated by Western blot and RT-qPCR analysis (Figure 1C,D). To further elucidate the potential involvement of GSR in pulmonary fibrosis, we examined its response to the differentiation of lung fibroblasts into myofibroblasts induced by TGF-β. A marked downregulation of GSR protein and mRNA levels in lung fibroblasts treated with TGF-β (Figure 1E,F). Additionally, immunofluorescence assays confirmed a substantial decrease in GSR protein expression in MRC5 cells following TGF-β administration (Appendix A).

Next, we developed a BLM-induced lung fibrosis mouse model to further examine this observation (Figure 1G). Consistent with our findings in human samples, both IHC, Western blot, and RT-qPCR analyses revealed a significant decrease in GSR expression in the lungs of BLM-treated animals compared to those receiving saline injections (Figure 1H–J). Similarly, in BLM-treated MLE-12 cells and TGF-β-induced PMLFs, GSR was also significantly reduced at both protein and mRNA levels (Figure 1K–N). These findings collectively suggest that GSR was downregulated in lung fibrosis and might play a critical role in the pathogenesis of pulmonary fibrosis.

### 3.2. Downregulation of GSR Promotes the Migration, EMT Process, and Senescence of A549 Cells

To further elucidate the impact of GSR on the migratory capabilities of A549 cells, we conducted both transwell and cell scratch assays. In the cell scratch test, it was evident that the migratory potential of A549 cells was markedly enhanced post-GSR interference (Figure 2A,B). Similarly, the transwell assay revealed that reducing GSR levels facilitated the migratory abilities of these cells (Figure 2C,D). Moreover, Western blot analyses demonstrated a significant decline in E-cadherin expression along with increases in N-cadherin and vimentin expression upon GSR knockdown in A549 cells (Figure 2E,F), thereby suggesting that GSR interference accelerates the EMT process within these cells. We also found a similar phenomenon in MLE-12 cells (Appendix A).

Cellular senescence is a major contributor to epithelial cell dysfunction and holds a pivotal role in the pathogenesis of IPF. GSR interference led to an increase in the expression of P53 and P21 proteins in A549 cells (Figure 2G,H) and MLE-12 cells (Appendix A). Additionally, β-galactosidase staining experiments demonstrated an augmented level of cell senescence following GSR interference (Figure 2I,J). Collectively, these findings underscore the notion that reduced GSR expression facilitates the migration, EMT process, and aging of A549 cells.

### 3.3. GSR Attenuates Migration, Senescence, and EMT Phenotypes in A549 Cells via Modulating GSH Levels

In our investigation into the regulatory role of GSR in maintaining cellular redox homeostasis in A549 cells, we observed that GSR manipulation resulted in elevated ROS levels as evidenced by fluorescence detection (Figure 3A). Additionally, using a malondialdehyde (MDA) assay, we noted an increase in lipid peroxidation levels post-GSR intervention (Figure 3B), indicating that such modulation induced a state of oxidative stress in the cells. Furthermore, when examining the effects of GSR interference on GSH levels, we found a significant reduction in GSH content compared to controls (Figure 3C). This suggests that GSR deficiency increased ROS accumulation via a decrease in GSH concentrations.

Additionally, we discovered that the elevation in ROS levels could be mitigated through the supplementation of exogenous GSH (Figure 3D,E). Following GSR interference in A549 cells, the addition of exogenous GSH was found to counteract the augmented migratory capability induced by this interference (Figure 3F,G). Moreover, Western blot analyses revealed that exogenous GSH not only enhanced the levels of E-cadherin but also suppressed the expressions of N-cadherin and vimentin (Figure 3H,I). These findings collectively demonstrate that the introduction of exogenous GSH attenuates the EMT process exacerbated by diminished GSR expression.

Following the intervention of GSR, we supplemented with GSH to examine its effects on cell senescence. Addition of GSH led to a decrease in the protein levels of P53 and P21 (Figure 3J,K). Compared to GSR intervention alone, the application of exogenous GSH resulted in a reduction of cell senescence following GSR silencing (Figure 3L,M). These findings suggest that diminished GSR expression enhances A549 cell senescence, whereas the inclusion of extrinsic GSH can mitigate this senescence phenotype.

### 3.4. Low Expression of GSR Promoted the Migration, Activation, and Senescence of Lung Fibroblasts

Subsequently, we delved deeper into the impact of GSR on MRC5 cells. Notably, the expression levels of FN1, COL1A1, and α-SMA were markedly elevated following GSR interference (Figure 4A,B). In addition, silencing GSR in PHLFs also increased the COL1A1 expression (Appendix A). A substantial enhancement in the migratory capability of MRC5 cells (Figure 4C,D) and a pronounced increase in the positive staining of β-galactosidase in MRC5 cells post-GSR intervention (Figure 4E,F) were observed. Additionally, the age-associated proteins P53 and P21 exhibited a significant upsurge in their expression levels following GSR modulation (Figure 4G,H). Collectively, these findings suggest that the depletion of GSR fosters the activation, migration, and senescence of MRC5 cells.

### 3.5. Activation of the Fibroblast Was Dependent on the GSH

We next investigated the role of GSR in modulating the redox status in lung fibroblasts. The same elevation in the ROS (Figure 5A,B) and MDA levels (Figure 5C) and a reduction in intracellular GSH levels (Figure 5D) were found in the MRC5 cells following the ablation of GSR. Supplementing with GSH significantly mitigated the ROS levels (Figure 5E,F), the FN1, COL1A1, and α-SMA expressions (Figure 5G,H), and migratory capacity (Figure 5I,J) of MRC5 cells, which were elevated by GSR deficiency.

Additionally, we discovered that the administration of exogenous GSH attenuated the elevation of P53 and P21 levels (Figure 5K,L). Moreover, β-galactosidase staining assays revealed a pronounced increase in β-galactosidase-positive staining following GSR interference, which was reversed by the addition of exogenous GSH (Figure 5M,N). These findings collectively suggest that the downregulation of GSR facilitates the migration, activation, and senescence of lung fibroblasts through the modulation of GSH levels.

### 3.6. GSR Inhibited the Epithelial Cell-Derived Fibroblast Activation Through TGF-β/Smad2 Signaling Pathway

To further elucidate the role of GSR in pulmonary fibrosis, we posited that GSR might contribute to abnormal AECII injury and repair, potentially leading to dysregulated interactions between AECII and mesenchymal cells. To test this hypothesis, we downregulated GSR expression in A549 cells and co-cultured them with MRC5 cells. Western blot analysis revealed an increase in the expression of COL1A1, TGF-β, and α-SMA in MRC5 cells (Figure 6A,B).

We hypothesized that interfering with GSR in A549 cells activates MRC5 cells through the secretion of TGF-β. The results showed that GSR interference significantly promoted the release of TGF-β in A549 cells (Figure 6C). Western blot analysis also showed that GSR intervention activated the TGF-β/Smad2 signaling pathway in A549 cells (Figure 6D). In addition, we examined the TGF-β/Smad2 signaling pathway in MRC5 cells following GSR interference. Western blot analysis revealed a significant increase in the p-Smad2 protein levels after GSR intervention, indicating that the TGF-β/Smad2 signaling pathway is indeed activated in MRC5 cells (Figure 6F,G). To investigate whether ROS accumulation resulting from GSR knockdown activates the TGF-β/Smad2 signaling pathway, we treated A549 and MRC-5 cells with NAC. Intracellular ROS measurement confirmed that NAC reversed the increase in ROS levels induced by GSR knockdown (Figure 6H–K). Furthermore, NAC treatment also inhibited the activation of the TGF-β/Smad2 signaling pathway triggered by GSR knockdown (Figure 6L–O). Our findings demonstrate that GSR interference in A549 cells produces TGF-β, which subsequently activates MRC5 cells by triggering the classical TGF-β/Smad2 signaling pathway.

## 4. Discussion

This study identified decreased GSR expression in both IPF patients and BLM-treated murine models, prompting investigation into its functional role in pulmonary fibrosis. Genetic disruption of GSR in key lung cell populations—specifically AECIIs and fibroblasts—resulted in reduced GSH levels, elevated ROS, activation of the TGF-β/Smad2 signaling pathway, and consequent promotion of cellular phenotypes, including enhanced migration, EMT, cellular senescence, and fibroblast activation. These findings suggest that GSR may be essential in reducing the fibrotic progression of IPF.

IPF is a progressive fibrotic disorder characterized by high morbidity and mortality rates. EMT represents an early phase in physiological wound repair during lung fibrosis [33]. EMT is typically identified histologically by the lack of epithelial cell–cell adhesion molecules, such as E-cadherin, and the emergence of mesenchymal markers, including N-cadherin [34]. Our study revealed that reducing GSR levels enhances the EMT of AECII when examining relevant markers.

In wound healing, fibroblasts become myofibroblasts with elevated α-SMA expression, vital for wound closure and collagen deposition [35]. In IPF, increased myofibroblasts drive fibrosis. Reducing GSR boosts fibrogenic gene expression and activates lung fibroblasts, an effect counteracted by added GSH.

The likelihood of developing idiopathic pulmonary fibrosis escalates markedly with advancing age, exhibiting a doubling of incidence approximately every ten years beyond the age of fifty [36]. Senescence stops cell growth and reduces replication, making lungs prone to fibrosis by blocking alveolar precursor cell renewal and creating a fibrogenic environment. In IPF, AECII shows aging traits, and single-cell RNA sequencing shows high senescent protein transcript levels in abnormal epithelial cells, mainly in fibroblastic foci. [37,38].

Oxidative injury, which occurs because of aging, is increased in IPF [39,40]. The BLM-induced pulmonary fibrosis model is a vital research tool for investigating the initiation and development of pulmonary fibrosis [41,42]. Previous studies indicate that exposure of alveolar epithelial cells (AECs) to BLM induces oxidative stress through ROS, leading to their eventual demise [43]. In this investigation, suppressing GSR expression led to a substantial rise in ROS concentrations within lung epithelial cells, paving the way for ROS-mediated injury to these cells and initiating pulmonary fibrosis. Oxidative stress occurs when excess reactive species such as ROS overwhelm antioxidant defenses, damaging cellular components like lipids, proteins, and nucleic acids [44]. The disorder of redox balance is considered the pathogenic factor of pulmonary fibrosis [45], and the regulatory mechanism of redox in the development of pulmonary fibrosis is still largely unknown. GSR catalyzes the production of glutathione, which is a crucial part of a cell’s antioxidant defense system and helps manage oxidative stress [46]. Our study demonstrated that disrupting GSR function markedly diminished the antioxidant capabilities of AECII, leading to elevated intracellular ROS levels and a pronounced upsurge in aging-associated protein expression. In addition, GSR knockdown impaired mitochondrial membrane potential, indicating compromised mitochondrial function. However, NAC-mediated ROS scavenging restored functional integrity, demonstrating that ROS accumulation underlies GSR deficiency-induced mitochondrial dysfunction (Appendix A). Analogous to epithelial cells, deregulated cellular senescence also manifests in IPF fibroblasts [34]. The findings suggest that reduced GSR expression could enhance the activation of pulmonary fibrosis by elevating ROS levels as part of the aging process.

Upon epithelial cell damage, the lung initiates self-repair via the wound-healing mechanism akin to other organs. Normal wound healing entails recruiting fibroblasts, constructing the extracellular matrix, and generating myofibroblasts responsible for synthesizing collagen and exerting force to seal the wound [34]. To delve deeper into the role of GSR in pulmonary fibrosis, we established an AECII-fibroblast coculture system. TGF-β signaling is well-established as a key player in pulmonary fibrosis, and its involvement in IPF has been comprehensively discussed in the previous literature [32,47]. Surprisingly, our study revealed that silencing GSR in AECII triggers the TGF-β/Smad2 signaling pathway in lung fibroblasts, offering new insights into GSR’s anti-fibrotic properties.

The findings demonstrate that the lack of GSR resulted in EMT progression in A549 cells and promoted the activation of MRC5 cells. Knockdown of GSR enhanced migratory capacity and induced cellular senescence in both A549 and MRC5 cells. Conversely, GSH, the product of GSR’s catalytic activity, mitigated these effects. Mechanistically, GSR disruption boosts intracellular ROS levels and then activates the TGF-β/Smad2 pathway, shedding light on the key mechanism in oxidative stress-induced pulmonary fibrosis and providing new understanding for managing age-related diseases.

## 5. Conclusions

Our study demonstrated that significant downregulation of GSR expression leads to EMT progression in A549 cells and promotes the activation of MRC5 cells. Genetic perturbation of GSR enhanced migratory capacity and induced cellular senescence in both A549 and MRC5 cells. Mechanistically, GSR silencing depleted intracellular GSH and then triggered ROS accumulation. This ROS surge subsequently activated the TGF-β/Smad2 signaling pathway, ultimately driving pathogenic processes underlying IPF.

In summary, GSR deficiency exacerbates and promotes pulmonary fibrosis by inhibiting GSH levels and increasing the accumulation of ROS, thereby activating the TGF-β/Smad2 signaling pathway, suggesting GSR is a potential therapeutic target for IPF.

## Figures and Tables

**Figure 1 biomolecules-15-01050-f001:**
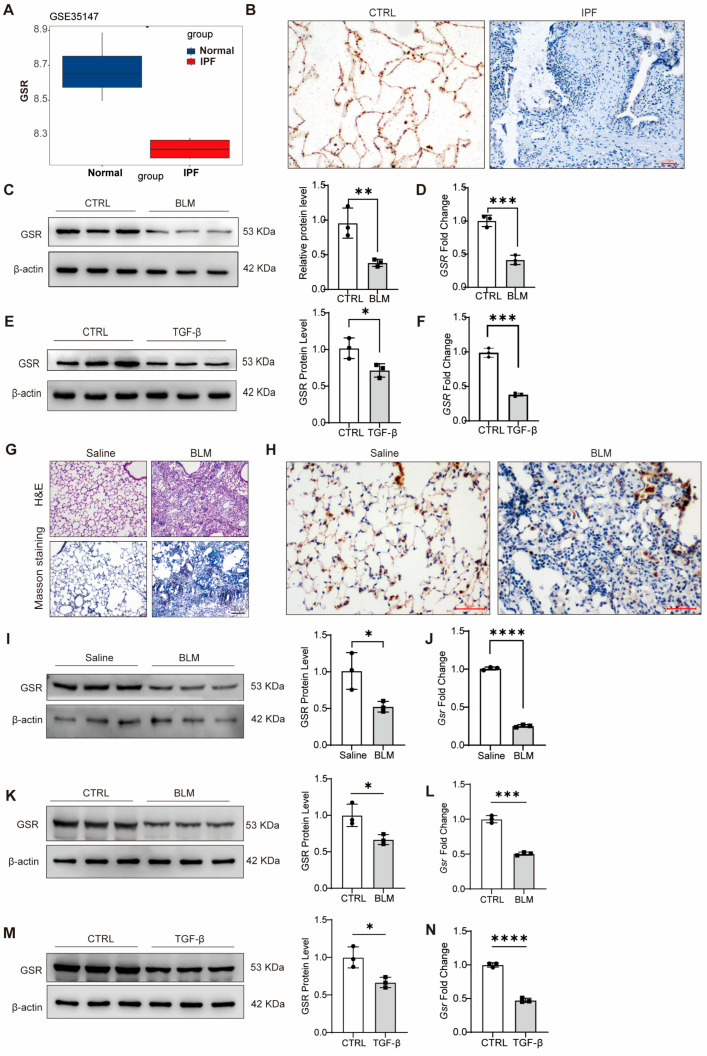
GSR was downregulated in lung fibrosis. (**A**) Box plot displays the representative gene GSR differentially expressed in Normal and IPF. * *p* < 0.05, Wilcoxon rank-sum test. (**B**) IHC was used to test the expression of GSR in IPF. Bar = 50 μM. (**C**) A549 cells were treated with BLM for 48 h. Western blot showed that the GSR protein level was downregulated. The right is its quantitative statistics. *n* = 3. (**D**) A549 cells were treated with BLM for 48 h. RT-qPCR analysis showed that the GSR mRNA level was downregulated. *n* = 3. (**E**) MRC5 cells were treated with TGF-β for 48 h. Western blot results showed that the protein level of GSR was significantly reduced under the action of TGF-β. *n* = 3. (**F**) MRC5 cells were treated with TGF-β for 48 h. RT-qPCR results showed that the mRNA level of GSR was significantly reduced under the action of TGF-β. The right is its quantitative statistics. *n* = 3. (**G**) HE and Masson staining were used to verify the successful establishment of a mouse pulmonary fibrosis model. (**H**) IHC was used to test the expression of GSR in the BLM-induced lung fibrosis mouse model. Bar = 50 μM. (**I**) Western blot analysis showed that GSR expression was downregulated in BLM-treated mice compared to saline-treated mice. The right is its quantitative statistics. *n* = 3. (**J**) RT-qPCR analysis showed that Gsr expression was downregulated in BLM-treated mice compared to saline-treated mice. *n* = 3. (**K**) MLE-12 cells were treated with BLM for 48 h. Western blot results showed that the protein level of GSR was significantly reduced under the action of BLM for 48 h. *n* = 3. (**L**) MLE-12 cells were treated with BLM for 48 h. RT-qPCR results showed that the mRNA level of Gsr was downregulated. (**M**) PMLFs were treated with TGF-β for 48 h. Western blot results showed that the protein level of GSR was significantly reduced under the action of TGF-β. *n* = 3. (**N**) PMLFs were treated with TGF-β for 48 h. RT-qPCR results showed that the mRNA level of Gsr was significantly reduced under the action of TGF-β. The right is its quantitative statistics. *n* = 3. Data are presented as mean ± SD. * *p* < 0.05, ** *p* < 0.01, *** *p* < 0.001, and **** *p* < 0.0001. Original WB images are provided in Appendix A.

**Figure 2 biomolecules-15-01050-f002:**
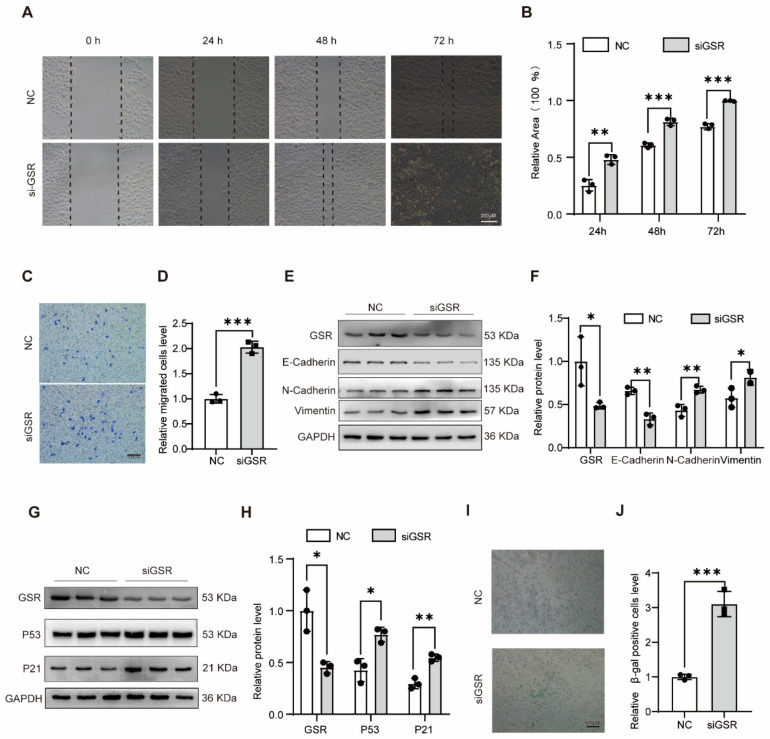
Downregulation of GSR, which promoted the migration, EMT process, and aging of A549 cells. (**A**,**B**) Wound healing assays were conducted and photographed at 0, 24, 48, and 72 h, with quantification (*n* = 3). Bar = 200 μM. (**C**,**D**) The changes in cell migration ability after interfering with GSR in A549 cells were detected by the transwell experiment, and the results were statistically analyzed. *n* = 3. (**E**) The expression of E-cadherin, N-cadherin, vimentin and GAPDH was detected by Western blot. (**F**) The quantitative statistics of (**E**). *n* = 3. (**G**,**H**) Western blot was used to detect the changes in senescence-related genes P53 and P21 in A549 cells after GSR intervention and their gray level analysis. *n* = 3. (**I**,**J**) β-galactosidase staining was used to detect the changes in cell aging in A549 cells after treating with GSR small interfering RNA (siGSR) and its statistical analysis diagram. *n* = 3. Data are presented as mean±SD. * *p* < 0.05, ** *p* < 0.01, and *** *p* < 0.001. Original WB images are provided in Appendix A.

**Figure 3 biomolecules-15-01050-f003:**
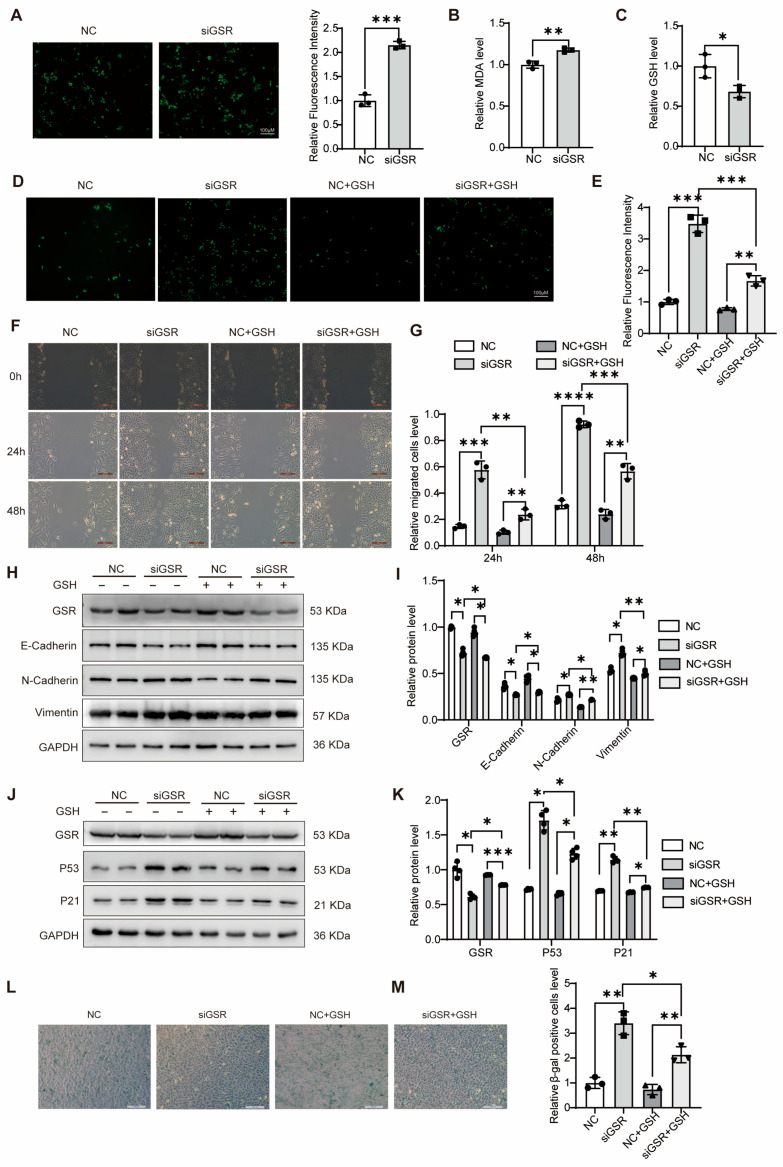
GSR alleviated migration, senescence, and EMT phenotypes of A549 cells by regulating GSH levels. (**A**). A ROS detection kit was used to detect ROS after siGSR in A549 cells. *n* = 3. (**B**) The MDA kit was used to detect the changes in lipid peroxidation levels in A549 cells. *n* = 3. (**C**) The changes in GSH after siGSR in A549 cells. *n* = 3. (**D**,**E**) After interfering with GSR for 24 h, GSH was added for 24 h, and then we detected the changes in intracellular ROS. *n* = 3. (**F**,**G**) After interfering with GSR in A549 cells, GSH was added, and the pictures taken by the inverted microscope through the scratch experiment and the statistical analysis of the results showed that the results were expressed as the percentage of wound healing. *n* = 3. Bar = 200 μM. (**H**,**I**) After interfering with GSR in A549 cells for 24 h, GSH was added for 24 h. The expression changes in EMT-related genes, E-cadherin, N-cadherin, and vimentin, were detected by Western blot, and then conduct statistical analysis on the results. *n* = 4. (**J**,**K**) After adding GSH to siGSR in A549 cells, the changes in P53 and P21 were detected by Western blot, and then conduct statistical analysis on the results. *n* = 4. (**L**,**M**) After interfering with GSR for 24 h, GSH was added and processed for 24 h, and then we detected the changes in cell senescence by β-galactosidase staining. *n* = 3. Bar = 200 μM. Data are presented as mean ± SD. * *p* < 0.05, ** *p* < 0.01, *** *p* < 0.001, and **** *p* < 0.0001. Original WB images are provided in Appendix A.

**Figure 4 biomolecules-15-01050-f004:**
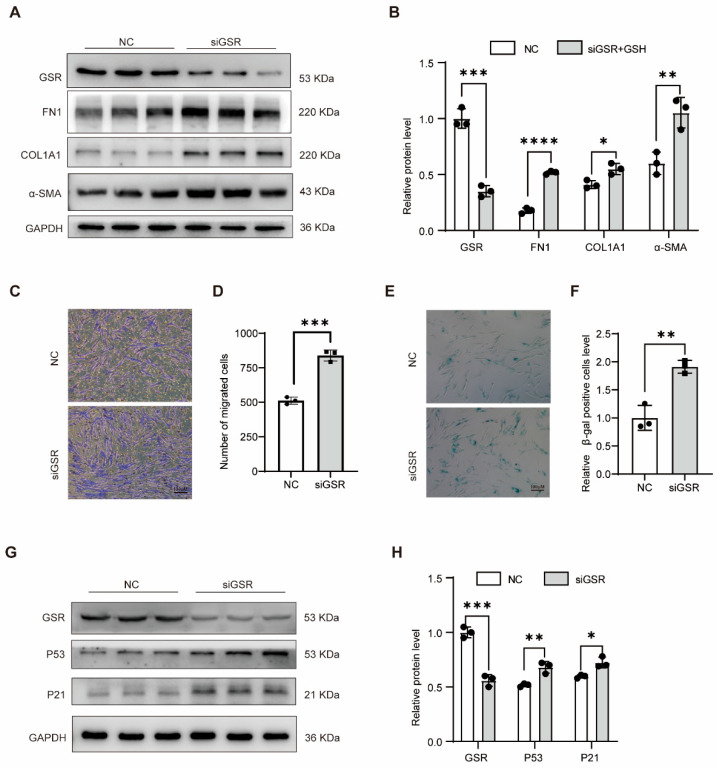
Downregulation of GSR promoted the activation, migration, and aging of MRC5 cells. (**A**,**B**) Western blot is used to detect the expression change in profibrotic gene protein levels after interfering with GSR in MRC5 cells processed for 48 h and its quantitative statistical diagram. *n* = 3 (**C**,**D**) The transwell experiment was used to detect the changes in MRC5 cells’ migration ability after interfering with GSR in MRC5 cells for 48 h and its quantitative statistical diagram. *n* = 3. (**E**,**F**) β-galactosidase staining was used to detect the changes in cell aging after siGSR in MRC5 cells and its statistical analysis diagram. *n* = 3. (**G**,**H**) Western blot was used to detect the protein levels of P53 and P21 in MRC5 cells after GSR intervention in MRC5 cells and their gray level analysis. *n* = 3. Data are presented as mean ± SD. * *p* < 0.05, ** *p* < 0.01, *** *p* < 0.001, and **** *p* < 0.0001. Original WB images are provided in Appendix A.

**Figure 5 biomolecules-15-01050-f005:**
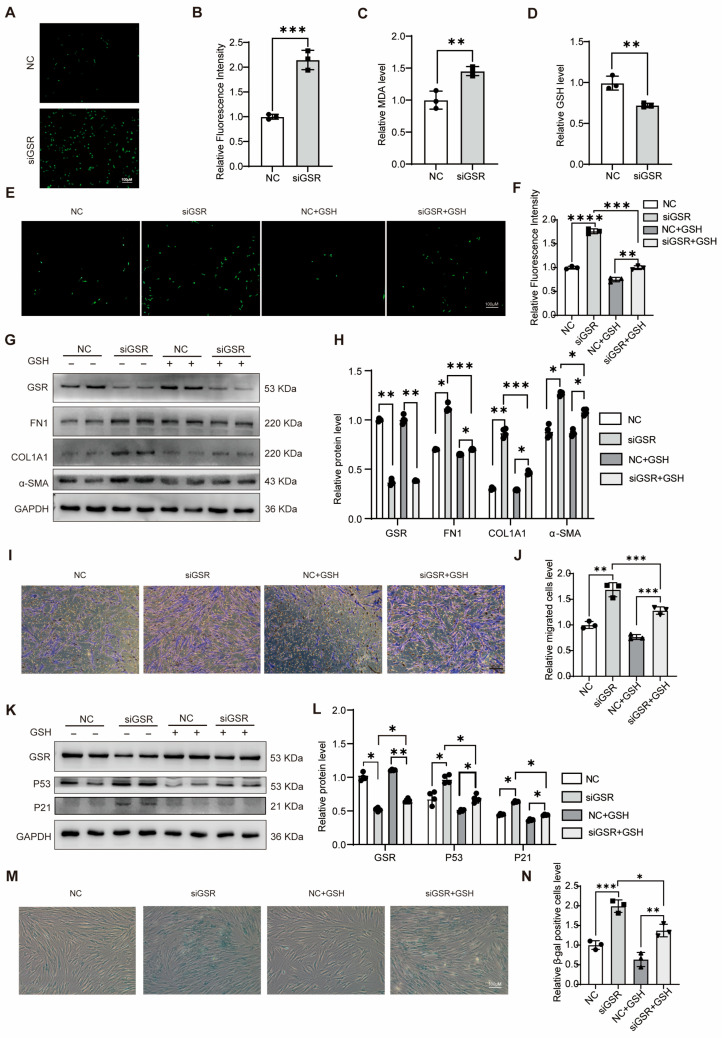
GSR affected the fibrotic phenotype of MRC5 cells by regulating the level of GSH. (**A**,**B**) ROS detection kit was used to detect ROS after siGSR in MRC5 cells. *n* = 3. (**C**) An MDA kit was used to detect the changes in lipid peroxidation levels in cells. *n* = 3. (**D**) The level of GSH in MRC5 cells after siGSR. *n* = 3. (**E**,**F**) The changes in intracellular ROS. *n* = 3. (**G**,**H**) Western blots of GSR, FN1, COL1A1, α-SMA, and GAPDH in MRC5 cells and the statistical diagram of gray analysis were obtained. *n* = 4. (**I**) Transwell migration assay of MRC5 cells at 36 h (*n* = 3). Bar = 100 μm. (**J**) Transwell experiments quantitative statistical analysis. *n* = 3. (**K**,**L**) Adding GSH after siGSR in MRC5 cells, detecting the changes in P53 and P21 by Western blot, and analyzing the statistical diagram of gray level. *n* = 4. (**M**,**N**) β-galactosidase staining to detect the changes in cell senescence. *n* = 3. Data are presented as mean ± SD. * *p* < 0.05, ** *p* < 0.01, *** *p* < 0.001, and **** *p* < 0.0001. Original WB images are provided in Appendix A.

**Figure 6 biomolecules-15-01050-f006:**
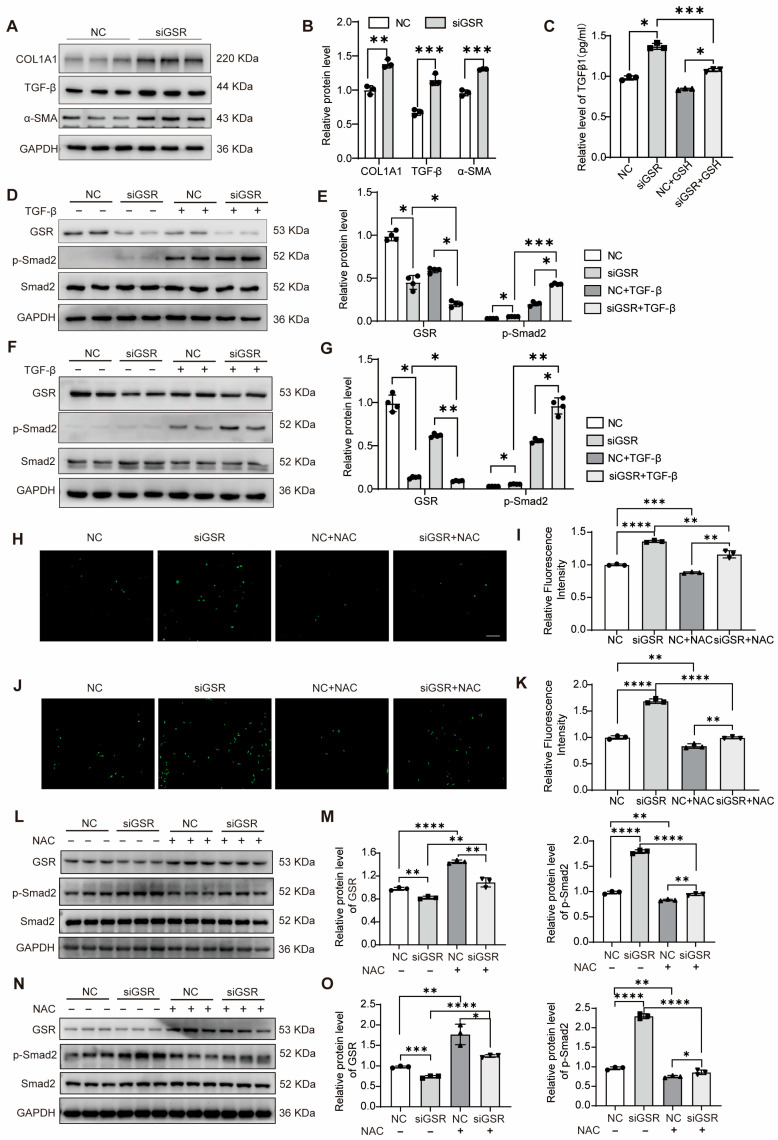
Silencing of GSR activates the TGF-β/smad2 signaling pathway, affecting cell phenotype. (**A**,**B**) After interfering with GSR in A549 for 48 h, the supernatant was taken to treat MRC5 cells for another 48 h, and protein levels of COL1A1, TGF-β, and α-SMA in MRC5 cells were detected by Western blot. The right is its quantitative statistics. *n* = 3. (**C**) The ELISA kit detects the change in TGF-β content. *n* = 3. (**D**,**E**) Adding TGF-β for 24 h after interfering with GSR in A549 cells for 24 h, thereby using Western blot to detect the influence of GSR on the TGF-β/Smad2 signal pathway. The right is its quantitative statistics. *n* = 4. (**F**,**G**) Adding TGF-β for 24 h after interfering with GSR in MRC5 cells for 24 h, thereby using Western blot to detect the influence of GSR on the TGF-β/Smad2 signal pathway. The right is its quantitative statistics. *n* = 4. (**H**) The intracellular ROS changes in A549 cells. *n* = 3. Bar = 50 μM. (**I**) The quantitative statistics of (**H**). (**J**) The intracellular ROS changes in MRC5 cells. *n* = 3. Bar = 100 μM. (**K**) The quantitative statistics of (**J**). (**L**) Treating with NAC for 24 h after interfering with GSR in A549 cells for 24 h, thereby using Western blot to detect the influence of GSR on the TGF-β/Smad2 signal pathway. (**M**) Quantitative statistics of (**L**). *n* = 3. (**N**) Treating with NAC for 24 h after interfering with GSR in MRC5 cells for 24 h, thereby using Western blot to detect the influence of GSR on the TGF-β/Smad2 signal pathway. (**O**) Quantitative statistics of (**N**). *n* = 3. Data are presented as mean ± SD. * *p* < 0.05, ** *p* < 0.01, *** *p* < 0.001, and **** *p* < 0.0001. Original WB images are provided in Appendix A.

## Data Availability

The data that support the findings of this study are available from the corresponding author, Lan Wang, upon reasonable request.

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
