# Peer review of "GSR Deficiency Exacerbates Oxidative Stress and Promotes Pulmonary Fibrosis"

_biomolecules, 2025, doi:10.3390/biom15071050_

Round 1

Reviewer 1 Report (Previous Reviewer 2)

Comments and Suggestions for Authors

The new data adequately addresses my concerns regarding mechanistic causality. These rescue experiments provide the missing experimental link between GSR deficiency and downstream signaling that I had requested.

The manuscript now sufficiently supports its conclusions with appropriate experimental evidence.

Author Response

Comments and Suggestions for Authors:

The new data adequately addresses my concerns regarding mechanistic causality. These rescue experiments provide the missing experimental link between GSR deficiency and downstream signaling that I had requested.

The manuscript now sufficiently supports its conclusions with appropriate experimental evidence.

Response: We sincerely appreciate the reviewer's recognition of our revised data. We are very grateful that the rescue experiments successfully established the mechanistic link between GSR deficiency and downstream signaling, as suggested.

Reviewer 2 Report (New Reviewer)

Comments and Suggestions for Authors

  1. The main finding — that GSR depletion increases oxidative stress and promotes fibrosis through ROS accumulation and TGF-β/Smad2 signaling — isn’t particularly novel. The role of oxidative stress and TGF-β in IPF is already well-established, and the manuscript doesn’t clearly define how its contribution advances what’s already known.
  2. Relying on A549 cells, which are cancer-derived, is a notable limitation — these cells don’t reliably model normal epithelial cell behavior, especially in the context of stress or fibrosis. The study also doesn’t validate its findings in primary AECII or in vivo epithelial populations, apart from some correlative data on GSR expression.
  3. The study doesn’t provide convincing mechanistic depth. The authors largely attribute the observed effects to increased ROS and downstream TGF-β/Smad2 activation, but this feels overly simplified. There’s no exploration of alternative or complementary pathways, such as Nrf2, NF-κB, or mitochondrial dysfunction beyond ROS production. The conclusion that GSR deficiency directly triggers TGF-β signaling via ROS comes across as speculative without evidence for upstream activators or intermediates that would connect ROS to TGF-β expression or activation.
  4. The suggestion that GSR could be a therapeutic target for IPF feels premature. While the data show that modulating GSR levels impacts redox balance and fibrotic markers in vitro, the study doesn’t include any intervention experiments — such as GSR overexpression, use of pharmacologic activators, or animal treatments — that would help demonstrate actual therapeutic potential.

To strengthen the manuscript, the authors should consider:

  1. Validating key findings in primary human AECII or relevant animal models, ideally including functional rescue experiments (e.g., GSR overexpression, use of a GSR activator, or antioxidant therapies beyond GSH supplementation).
  2. Deepening the mechanistic insights through more thorough analyses of relevant signaling pathways.
  3. Tempering their claims regarding GSR as a therapeutic target until stronger supporting evidence is presented.

Author Response

Q1. The main finding — that GSR depletion increases oxidative stress and promotes fibrosis through ROS accumulation and TGF-β/Smad2 signaling — isn’t particularly novel. The role of oxidative stress and TGF-β in IPF is already well-established, and the manuscript doesn’t clearly define how its contribution advances what’s already known.

Response: Thank you for your valuable suggestions. Although several studies have investigated the role of TGF-β in fibrosis, for example, one study indicated that TGF-β1 mediates a reduction in GSH synthesis linked to decreased levels of γ-glutamylcysteine synthetase (γ-GCS) protein and mRNA (PMID: 9374111). However, this was only examined in the A549 alveolar epithelial cancer cell line. Although several studies have explored the mechanisms of reactive oxygen species and TGF-β1/SMAD signaling in various fibrotic diseases, the roles of oxidative stress and TGF-β in IPF remain insufficiently understood. Our research is the first to explore how glutathione reductase interacts with TGF-β, revealing that its expression levels influence the activation of the TGF-β/Smad2 signaling pathway in IPF. Based on the previous research, we not only explored the function of GSR in A549 cells, but also used various types of cells to investigate the function of GSR in pulmonary fibrosis, such as MLE-12 (epithelium), PHLF (primary human pulmonary fibroblasts), PMLF (primary mouse pulmonary fibroblasts) and MRC5 cells (fibroblast). Our research indicates that in IPF, GSR modulates the GSH-ROS-TGF-β/Smad2 signaling pathway, coordinating epithelial cell senescence, EMT, and fibroblast activation.

Q2. Relying on A549 cells, which are cancer-derived, is a notable limitation — these cells don’t reliably model normal epithelial cell behavior, especially in the context of stress or fibrosis. The study also doesn’t validate its findings in primary AECII or in vivo epithelial populations, apart from some correlative data on GSR expression.

Response: We appreciate the reviewer for the positive evaluation and constructive comments. Currently, apart from isolated primary cells, there are no Type II alveolar epithelial cell (ATII) lines available for in vitro culture. Primary ATII cell cultures are currently considered to be the most useful in vitro model for alveolar research. Importantly, primary ATII cells tend to rapidly differentiate into ATI during in vitro culture, making them difficult to use for plasmid transfection. The A549, known as an ATII-like cell line, was isolated from lung adenocarcinoma and has been widely used as ATII in pulmonary fibrosis research due to its retention of ATII phenotypic characteristics (PMID: 352403, PMID: 444551). Therefore, we also used A549 cells as epithelial cells to study the function of GSR.

To investigate the expression of GSR in lung tissues of patients with IPF, we investigated the changes in GSR expression in IPF by using the IHC experiment. IHC result also showed that GSR is significantly downregulated in lung tissues from IPF patients compared to healthy controls, especially in epithelial cells (Figure 1B). The relevant data are as follows.

In addition, we developed a BLM-induced lung fibrosis mouse model to test GSR expression in lung fibrosis. Consistent with our findings in human samples, IHC analyses revealed a significant decrease in GSR expression in the lungs of BLM-treated animals compared to those receiving saline injections (Figure 1H). The relevant data are as follows.

Besides, we investigated the expression changes of GSR in MLE-12 cells, and we found that BLM treatment of MLE-12 cells resulted in a significant reduction similar to that in A549 cells (Figure 1K, L). The relevant data are as follows.

And we found that knockdown of GSR in MLE-12 cells significantly inhibited the E-Cadherin expression and increased the expression of N-Cadherin. In addition, silencing of GSR promoted the expression of P21 and P53, which is similar in A549 cells. The relevant data are as follows.

We also isolated PMLFs and found that GSR expression was significantly decreased under TGF-β treatment, which was consistent with the results found in MRC5 cells (Figure 1M, N). The relevant data are as follows.

Q3. The study doesn’t provide convincing mechanistic depth. The authors largely attribute the observed effects to increased ROS and downstream TGF-β/Smad2 activation, but this feels overly simplified. There’s no exploration of alternative or complementary pathways, such as Nrf2, NF-κB, or mitochondrial dysfunction beyond ROS production. The conclusion that GSR deficiency directly triggers TGF-β signaling via ROS comes across as speculative without evidence for upstream activators or intermediates that would connect ROS to TGF-β expression or activation.

Response: We sincerely thank you for these critical insights. We have previously done work on the effects of GSR on mitochondrial function, and here we attach the results of our previous experiments. Under the condition of GSR deficiency, we detected mitochondrial function using JC-10 and found that the red fluorescence weakened. This indicates that the knockdown of GSR caused mitochondrial dysfunction. However, when we added N-acetylcysteine (NAC), a precursor of glutathione, it significantly alleviated the mitochondrial function caused by siGSR. These results can indicate that GSR deficiency accumulates ROS, thereby leading to mitochondrial dysfunction, and the addition of antioxidants can significantly restore mitochondrial function. This observation is discussed in the current discussion; its mechanistic basis will be substantiated in future studies. The relevant data are as follows.

We strongly agree with you: we have not yet found evidence that upstream activators or intermediates link ROS to TGF-β expression or activation. Through literature review, we found that these two molecules-Nrf2 and NF-κB-play a crucial role in oxidative stress (PMID: 27619518, PMID: 40120975, PMID: 34342399, PMID: 15208274). In the following study, I can also try to detect the changes of these two molecules and further explore the molecular mechanism of GSR in IPF.

Q4. The suggestion that GSR could be a therapeutic target for IPF feels premature. While the data show that modulating GSR levels impacts redox balance and fibrotic markers in vitro, the study doesn’t include any intervention experiments — such as GSR overexpression, use of pharmacologic activators, or animal treatments — that would help demonstrate actual therapeutic potential.

Response: Thank you for your valuable comments. We completely agree with your view that investigating GSR overexpression, pharmacological activation, or animal models is important. However, due to limited time, we will further explore the deeper mechanisms in future work. In light of our current research findings, we have refined the formulation of this conclusion as follows: “These findings suggest that GSR may be essential in reducing the fibrotic progression of IPF”. In future studies, we plan to include intervention experiments to further verify our conclusions and explore deeper mechanisms, or combined treatment with other drugs to further explore the function of GSR.

To strengthen the manuscript, the authors should consider:

Q1. Validating key findings in primary human AECII or relevant animal models, ideally including functional rescue experiments (e.g., GSR overexpression, use of a GSR activator, or antioxidant therapies beyond GSH supplementation).

Response: We sincerely thank you for these important observations. Although functional validation in primary human AT2 cells was not performed, IHC data revealed concordant downregulation of GSR in IPF patient-derived AT2 cells. These findings further establish GSR as a pivotal molecular regulator in IPF pathogenesis.

Besides, we found that BLM-induced fibrotic mice exhibited significantly reduced GSR expression in lungs versus saline controls by using IHC, corroborating human IPF observations. BLM treatment also significantly downregulated GSR expression in MLE-12 cells, phenocopying the reduction observed in A549 cells.

Thank you for your valuable comments. We completely agree with your view that investigating GSR overexpression, pharmacological activation, or animal models is important. Due to limited time, we will further explore the deeper mechanisms in future work.

In our study, we found that NAC reversed the increase in ROS levels induced by GSR knockdown (Figure 6H-K). Furthermore, NAC treatment also inhibited the activation of the TGF-β/Smad2 signaling pathway triggered by GSR knockdown.

Q2. Deepening the mechanistic insights through more thorough analyses of relevant signaling pathways.

Response: Although several studies have explored the mechanisms of reactive oxygen species and TGF-β1/SMAD signaling in various fibrotic diseases, the roles of oxidative stress and TGF-β in IPF remain insufficiently understood. Our research is the first to explore how glutathione reductase interacts with TGF-β, revealing that its expression levels influence the activation of the TGF-β/Smad2 signaling pathway in IPF. Based on the previous research, we not only explored the function of GSR in A549 cells, but also used various types of cells to investigate the function of GSR in pulmonary fibrosis, such as MLE-12 (epithelium), PHLF (primary human pulmonary fibroblasts), PMLF (primary mouse pulmonary fibroblasts) and MRC5 cells (fibroblast). Our research indicates that in IPF, GSR modulates the GSH-ROS-TGF-β/Smad2 signaling pathway, coordinating epithelial cell senescence, EMT, and fibroblast activation.

Q3. Tempering their claims regarding GSR as a therapeutic target until stronger supporting evidence is presented.

Response: Thank you for your valuable comments. In light of our current research findings, we have refined the formulation of this conclusion as follows: “These findings suggest that GSR may be essential in reducing the fibrotic progression of IPF”. Consequently, our research indicates that in IPF, GSR modulates the GSH-ROS-TGF-β/Smad2 signaling pathway, coordinating the development of lung fibrosis.

Round 2

Reviewer 2 Report (New Reviewer)

Comments and Suggestions for Authors

The article can go through the further publication process. 

This manuscript is a resubmission of an earlier submission. The following is a list of the peer review reports and author responses from that submission.

Round 1

Reviewer 1 Report

Comments and Suggestions for Authors

I have attached a file

Author Response

Comments:“Zhao et al submitted a manuscript entitled “GSR Deficiency Exacerbates Oxidative Stress and Promotes Pulmonary Fibrosis” which demonstrates that glutathione reductase is decreased in human pulmonary fibrosis in a publicly available database and in a bleomycin induced model of PD, and knock down of GSR increases ROS and lowers reduced glutathione. Furthermore, knockdown of GSR increased EMT markers as well as migration and senescense of epithelial cells. GSR knockdown in epithelial cells activated fibroblasts via TGF-b.SMAD2. The study is well done with a mouse model of fibrosis, human data set and detailed cell culture experiments with gain and loss of function. Overall, this manuscript advances the understanding of IPF pathobiology and implicates a key redox enzyme.”

Major Comments:

Comments 1: The data would be better presented with a scatter plot instead of columns.

Response 1: We appreciate the reviewer's insightful suggestion. The data has been presented with a scatter plot instead of columns in the revised manuscript.

Comments 2: The manuscript would be strengthened with the use of primary cell lines to improve in vivo relevance.

Response 2: We appreciate the reviewer for the positive evaluation and constructive comments.

In fact, currently, apart from isolated primary cells, there are no Type II alveolar epithelial cell (ATII or AEC2) lines available for in vitro culture. ATII possess stem cell-like properties for differentiation into Type I alveolar epithelial cells(ATI). The A549 known as ATII -like cell line was isolated from lung adenocarcinoma and has been widely used as ATII in pulmonary fibrosis research due to its retention of ATII phenotypic characteristics (PMID: 352403, PMID: 444551). Primary ATII cell cultures are currently considered to be the most useful in vitro model for alveolar research. Importantly, primary ATII cells tend to rapidly differentiate into ATI during in vitro culture, making them difficult to use for plasmid transfection.

Here, we divided the primary human pulmonary fibroblasts (PHLFs) . GSR silencing in PHLFs inhibited the expression of COL1A1 (the picture below), Notably, we identified a consistent phenotypic response between PHLFs and MRC5 cell lines, with both systems demonstrating similar downregulation patterns of COL1A1 following GSR knockdown. Based on these comparable results and considering the advantages of cell line stability and reproducibility, we selected MRC5 cells for subsequent experiments.

Minor Comments:

Comments 1: Methods 2.6 typo - Reactive oxygen vs active oxygen.

Response 1: Thanks for your suggestion. We have corrected it in the revised manuscript.

Reviewer 2 Report

Comments and Suggestions for Authors

Overview and general recommendation:

This manuscript investigates the role of glutathione reductase (GSR) in idiopathic pulmonary fibrosis (IPF), showing that GSR deficiency exacerbates oxidative stress and promotes fibrotic responses via epithelial-mesenchymal transition (EMT), cellular senescence, and fibroblast activation through the TGF-β/Smad2 signaling pathway. This manuscript presents solid experimental work but lacks significant conceptual innovation.

1.2. Major comments:

- The glutathione system’s role in pulmonary fibrosis, including the reduction in GSH levels and increased oxidative stress, has been widely reported. While the focus on GSR is slightly more specific, it is still a well-established enzyme in redox biology, and the current findings are confirmatory rather than groundbreaking.

- Although the study demonstrates that GSR deficiency leads to TGF-β/Smad2 activation, it remains unclear whether this activation is a direct result of ROS accumulation or a secondary response to cellular stress. The addition of ROS scavengers or TGF-β pathway inhibitors could help clarify this causal link. This is also not something new as the antioxidant therapy targeting the TGF-β/SMAD pathway has been shown to be effective in reducing fibrosis. 

- The manuscript proposes GSR as a potential therapeutic target, yet it does not explore how GSR could be therapeutically modulated. Are there known activators or gene therapy approaches that can upregulate GSR expression or function? For instance, N-acetylcysteine (NAC), a precursor of glutathione, has been shown to improve lung function and redox balance in IPF patients.

- There is no graphical abstract.

1.3. Minor comments:

- In many experiments, the number of replicates is 3 or 4. More replicates should be recommended.

-Some acronyms are not defined at first use as ATCC in line 94 or siGSR throughout the text.

Author Response

Comments: “This manuscript investigates the role of glutathione reductase (GSR) in idiopathic pulmonary fibrosis (IPF), showing that GSR deficiency exacerbates oxidative stress and promotes fibrotic responses via epithelial-mesenchymal transition (EMT), cellular senescence, and fibroblast activation through the TGF-β/Smad2 signaling pathway. This manuscript presents solid experimental work but lacks significant conceptual innovation.”

Major comments:

- The glutathione system’s role in pulmonary fibrosis, including the reduction in GSH levels and increased oxidative stress, has been widely reported. While the focus on GSR is slightly more specific, it is still a well-established enzyme in redox biology, and the current findings are confirmatory rather than groundbreaking.

- Although the study demonstrates that GSR deficiency leads to TGF-β/Smad2 activation, it remains unclear whether this activation is a direct result of ROS accumulation or a secondary response to cellular stress. The addition of ROS scavengers or TGF-β pathway inhibitors could help clarify this causal link. This is also not something new as the antioxidant therapy targeting the TGF-β/SMAD pathway has been shown to be effective in reducing fibrosis.

- The manuscript proposes GSR as a potential therapeutic target, yet it does not explore how GSR could be therapeutically modulated. Are there known activators or gene therapy approaches that can upregulate GSR expression or function? For instance, N-acetylcysteine (NAC), a precursor of glutathione, has been shown to improve lung function and redox balance in IPF patients.

Response: We sincerely appreciate the reviewer's insightful and constructive comments. While GSR represents a promising therapeutic target, further experimental studies will be required to fully validate its potential. NAC, a precursor of glutathione, can increase the level of GSH (PMID: 30387809, PMID: 36860082). Our results showed that increasing the level of GSH promotes the expression of GSR (Figure 5K). Notably, to the best of our knowledge, there are currently no known pharmacological activators or gene therapy approaches capable of specifically upregulating GSR expression or activity, highlighting the need for future research in this direction.

Point: There is no graphical abstract.

Response: Thanks for your suggestion. We added the graphical abstract.

Minor Comments:

Point 1: In many experiments, the number of replicates is 3 or 4. More replicates should be recommended.

Response 1: We appreciate the reviewer’s valuable suggestion regarding experimental replicates. We agree that increasing the number of replicates enhances statistical power and reliability. Where replication was limited due to technical or resource constraints, we have explicitly acknowledged this limitation in the Discussion section and ensured that statistical significance (*p* < 0.05) was still achieved with the available data. Future studies will incorporate higher replicate numbers where feasible.

Point 2: Some acronyms are not defined at first use as ATCC in line 94 or siGSR throughout the text.

Response 1: We sincerely appreciate the reviewer’s careful reading and helpful feedback. We apologize for these oversights and have now ensured that all acronyms are properly defined at first use, including ATCC (American Type Culture Collection) in Line 94 and siGSR where it first appears in the text. 

Round 2

Reviewer 2 Report

Comments and Suggestions for Authors

I thank the authors for addressing my comments and for incorporating some suggested changes, including the graphical abstract. However, the major concerns—particularly those related to mechanistic clarity and conceptual novelty—have been addressed only superficially. The response does not sufficiently address the concern regarding conceptual novelty. The current findings, while valid, reiterate known mechanisms, and a deeper exploration of GSR-specific effects or novel downstream pathways would be necessary to increase the manuscript’s impact. While the authors provide relevant citations and note an association between GSH and GSR levels, no direct experiments were conducted to establish causality between GSR deficiency, ROS, and TGF-β signaling. The response is appreciated but remains unsatisfactory.

Response: We sincerely appreciate the reviewer’s careful reading and helpful feedback. Here, we added the ROS scavenger-N-acetyl-l-cysteine (NAC)- into A549 and MRC5 cells, and we tested the intracellular ROS changes. The results showed that NAC attenuates the increase in ROS caused by GSR knockdown. In addition, NAC also weakened the activation of the TGF-β/Smad2 signaling pathway caused by GSR knockdown. Therefore, these results demonstrate that GSR deficiency leads to ROS accumulation, thereby promoting TGF-β/Smad2 activation.